# Generation of Induced Pluripotent Stem Cells and Neuroepithelial Stem Cells from a Family with the Pathogenic Variant p.Q337X in Progranulin

**DOI:** 10.3390/ijms262311242

**Published:** 2025-11-21

**Authors:** Katarzyna Gaweda-Walerych, Adam Figarski, Sylwia Gawlik-Zawiślak, Marta Woźniak, Anna Chołoniewska, Natalia Mierzwa, Eliza Lutostańska, Jakub Szymanowski, Michalina Wężyk

**Affiliations:** 1Department of Neurogenetics and Functional Genomics, Mossakowski Medical Research Institute, Polish Academy of Sciences, 02-106 Warsaw, Poland; afigarski@imdik.pan.pl (A.F.); acholoniewska@imdik.pan.pl (A.C.); nmierzwa@imdik.pan.pl (N.M.); elutostanska@imdik.pan.pl (E.L.); 2Department of Genetics, Institute of Psychiatry and Neurology, Sobieskiego 9, 02-957 Warsaw, Poland; sgawlik@ipin.edu.pl (S.G.-Z.); mwozniak@ipin.edu.pl (M.W.); 3Laboratory of Advanced Microscopy Techniques (LAMT), Mossakowski Medical Research Institute, Polish Academy of Sciences, 02-106 Warsaw, Poland; jszymanowski@imdik.pan.pl

**Keywords:** progranulin, *GRN*, frontotemporal dementia (FTD), induced pluripotent stem cells (iPSCs), neuroepithelial stem cells (NES), pluripotency markers, embryoid body

## Abstract

Pathogenic *GRN* variants that reduce progranulin (PGRN) levels cause frontotemporal dementia (FTD). To facilitate model development, we generated induced pluripotent stem cells (iPSCs) from dermal fibroblasts of two family members carrying the *GRN* c.1009C>T (p.Q337X) pathogenic variant—one symptomatic and one asymptomatic—as well as a non-carrier first-degree relative serving as a genetically matched control. The obtained iPSC lines were validated for pluripotency markers (Nanog, Sox2, Oct4, and TRA1-1-81), genomic integrity, and differentiation potential. The obtained iPSC lines were subsequently directed toward neuroepithelial stem (NES) cells. NES identity was confirmed by the expression of lineage-specific markers, including Nestin and Sox2 (assessed by immunocytochemistry), as well as *SOX1*, *PLAGL1*, and *MKI67* (evaluated by real-time PCR). Furthermore, *GRN* mRNA levels were significantly reduced in iPSC and NES lines derived from mutation carriers compared to control cells. The established iPSC and NES cell lines represent a platform for modeling progranulin-deficient FTD. The symptomatic and asymptomatic carrier-derived lines obtained from the same family offer a unique opportunity to study disease progression across clinical phases. The control line, derived from a related (first-degree) non-carrier, minimizes genetic background variability. Their utility of the established cell lines extends to therapeutic drug screening and further differentiation into neuronal, non-neuronal, and organoid models.

## 1. Introduction

Heterozygous mutations in the *GRN* gene lead to reduced levels of progranulin (PGRN), causing frontotemporal dementia (FTD) characterized by frontotemporal lobar degeneration (FTLD) with TAR DNA-binding protein 43 (TDP-43) inclusions [1,2]. On the other hand, very rare homozygous *GRN* mutations lead to neuronal ceroid lipofuscinosis type 11 (NCL11)—a lysosomal storage disorder with onset in young adulthood, as reviewed by [3].

Frontotemporal lobar degeneration is the second most common form of early-onset dementia, typically affecting individuals under the age of 65 [4]. Heterozygous *GRN* mutations are found in approximately 13.9% of FTD patients [5]. These loss-of-function mutations lead to haploinsufficiency of progranulin, a protein involved in lysosomal function, neuronal survival, and neuroinflammation, as reviewed by [3].

One such pathogenic variant, rs1598364961 (c.1009C>T, p.Q337X, NM_002087.2), localizes to exon 9 of the *GRN* gene, which consists of 13 exons. This variant is listed as pathogenic in multiple databases, including ClinVar, and has been repeatedly identified in familial FTD cases since its initial description in 2007 [6,7,8,9]. In our previous work, we have characterized skin fibroblast lines derived from a symptomatic patient with familial FTD, carrying the p.Q337X mutation, alongside samples from much younger first-degree relatives: one asymptomatic also harboring the mutation, and a non-carrier serving as a control [10]. p.Q337X introduces a premature termination codon (PTC), triggering degradation of the aberrant transcript via nonsense-mediated mRNA decay (NMD) and leading to progranulin (PGRN) protein haploinsufficiency and lower mRNA in the carriers’ fibroblasts [10].

The current study builds upon these findings by generating and characterizing induced pluripotent stem cell (iPSC) lines and neuroepithelial stem cells (NES) from this family. To our knowledge, no patient-based models with the p.Q337X mutation have been developed so far [3]. We aim to establish a cellular model system to compare the effects of progranulin deficiency across disease stages (symptomatic vs. asymptomatic), while maintaining similar genetic backgrounds, as all three subjects are first-degree relatives.

## 2. Results

### 2.1. Characterization of iPSC Lines from a Family with GRN c.1009C>T (p.Q337X) Pathogenic Variant, and a Healthy First-Degree Relative

We generated iPSC lines from previously characterized dermal fibroblast lines [10] derived from a family comprising two members carrying *GRN* c.1009C>T (p.Q337X) pathogenic variant, a symptomatic one and an asymptomatic one, and a non-carrier first-degree relative. The patient designated as P1 was diagnosed with familial FTD; the subject P2 (two decades younger) remained asymptomatic (as of 2025). The ages of the subjects have been omitted to protect personal confidential information.

The obtained iPSC lines fulfilled the criteria listed in Table 1 and illustrated in Figure 1, Figure 2 and Figure 3.

The established lines formed colonies with the typical morphology of iPSCs, characterized by well-defined boundaries as visualized by Leica Integrated Modulation Contrast (IMC) (Figure 1a, right panel: the greyscale image below the line name). These colonies expressed key pluripotency markers, including Nanog, Sox2, Oct4, and TRA-1-81 (Figure 1a, left panels; the corresponding larger views, from which the highlighted rectangles were cropped, are presented in Appendix A). TRA-1-81 is an epitope on the Podocalyxin-like protein 1 (PODXL) expressed on the surface of pluripotent stem cells. The high expression of *NANOG* and *OCT4,* core pluripotency transcription factors, maintaining the undifferentiated state and self-renewal capacity of iPSC, was further validated at the mRNA level and contrasted with over 1000-fold lower expression of these markers in the corresponding fibroblast lines derived from P1, P2, and CTRL1 (Figure 1b, Table 1).

To assess the purity of the obtained iPSC lines (P1, P2, and CTRL1), we quantified Nanog- and DAPI-positive nuclei in fluorescence images acquired using a Zeiss Axioexaminer.Z1 microscope (Carl Zeiss Microscopy GmbH, Oberkochen, Germany). More than 18K cells were analyzed per line confirming the high purity of our colonies (see Table 2).

To verify the differentiation potential of the obtained iPSC lines into the three germ layers, the cells were subjected to in vitro embryoid body (EB) formation, followed by analysis of ectodermal, endodermal, and mesodermal markers (Figure 2, Table 1, Appendix A provides the corresponding split channels for detailed visualization). For ectoderm, β3-Tubulin (TUJ1) and Nestin were used. To better visualize the borders between the embryonic germ layers a co-staining using markers specific to both endo/ectodermal or meso/ectodermal lineages was used (Figure 2a). Endodermal marker Sox17 (SRY-Box Transcription Factor 17) was co-visualized with ectodermal marker Nestin, DAPI stained the nucleus. Among the endodermal markers, SOX17 is considered the most specific and reliable marker for definitive endoderm in embryoid bodies derived from iPSCs. SOX17 is expressed during the earliest stages of definitive endoderm formation and is not typically found in mesodermal or ectodermal lineages [11]. We also performed staining with NCAM1 (Neural Cell Adhesion Molecule 1) that is expressed in both neuroectoderm and paraxial mesoderm [12]. NCAM1 was visualized with ectodermal marker MAP2 (Microtubule-Associated Protein 2) to show the distinction between these two germ layers. Staining with CXCR4 (C-X-C chemokine receptor type 4) in EBs was performed to identify mesendodermal or cardiogenic mesoderm populations [13] (Figure 2a). CXCR4 was co-visualized with DAPI.

The differentiation potential of P1, P2, and CTRL1 iPSC lines into the three germ layers was assessed in parallel using the STEMdiff ™ Trilineage Differentiation Kit (STEMCELL Technologies, Vancouver, BC, Canada). Unlike the spontaneous and heterogeneous formation of 3D embryoid bodies, which typically requires over 21 days, this approach employs defined media to induce directed differentiation into ectoderm, mesoderm, and endoderm within 5–7 days. To confirm differentiation of P1, P2, and CTRL1 iPSC lines into a homogeneous mesodermal population, CXCR4 staining was performed as recommended in the protocol of the STEMdiff ™ Trilineage Differentiation Kit. CXCR4 staining revealed both membrane-bound and cytoplasmic localization (Appendix A). This pattern is consistent with CXCR4′s role as a membrane-bound G protein-coupled receptor (GPCR) that resides on the plasma membrane and undergoes constitutive endocytosis [13]. In addition, mRNA analyses showed significantly higher expression of *TBXT* mRNA, which encodes the T Brachyury transcription factor (a mesodermal marker), and increased expression of *FOXA2*, which encodes Forkhead Box A2 (an endodermal marker), in embryoid bodies (EBs) derived from P1, P2, and CTRL2, compared to their respective iPSC lines (Figure 2b).

**Figure 2 ijms-26-11242-f002:**
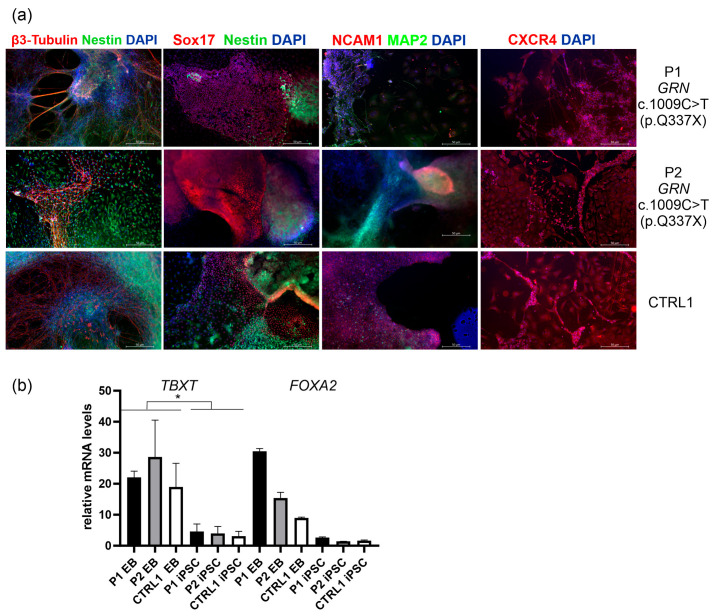
(**a**) In vitro differentiation of iPSCs into embryoid bodies (EB) and the analysis of the markers of the three germ layers (ectoderm, mesoderm and endoderm) by immunocytochemistry. For ectoderm, β3-Tubulin (TUJ1) (secondary antibody Alexa Flour 555 red) and Nestin (secondary antibody Alexa Flour 488 green) were used. To better visualize the borders between the embryonic germ layers a co-staining using markers specific to both endo/ectodermal or meso/ectodermal lineages was used. Endodermal marker Sox17 (SRY-Box Transcription Factor 17) was co-visualized with ectodermal marker Nestin (secondary antibody Alexa Flour 488 green), DAPI stained the nucleus (blue). NCAM1 (Neural Cell Adhesion Molecule 1) is expressed in both neuroectoderm and paraxial mesoderm [12]. NCAM1 (secondary antibody Alexa Flour 555 red) was visualized with ectodermal marker MAP2 (Microtubule-Associated Protein 2; secondary antibody Alexa Flour 488 green). CXCR4 (C-X-C chemokine receptor type 4) identifies mesendodermal or cardiogenic mesoderm populations [13]. CXCR4 was co-visualized with DAPI. Scale bar: 50 μm. The corresponding split images acquired from different fluorescence channels are provided in Appendix A. Original microscopy images are available in a single ZIP archive as Appendix A. (**b**) mRNA analyses in embryoid bodies (EBs) derived from P1, P2, and CTRL2 revealed significantly higher expression of *TBXT* mRNA (encoding the T Brachyury transcription factor, a mesodermal marker) and *FOXA2* mRNA (encoding Forkhead Box A2, an endodermal marker) compared to their respective iPSC lines. *p* was calculated in two-tailed *t*-test, * *p* < 0.05.

**Figure 3 ijms-26-11242-f003:**
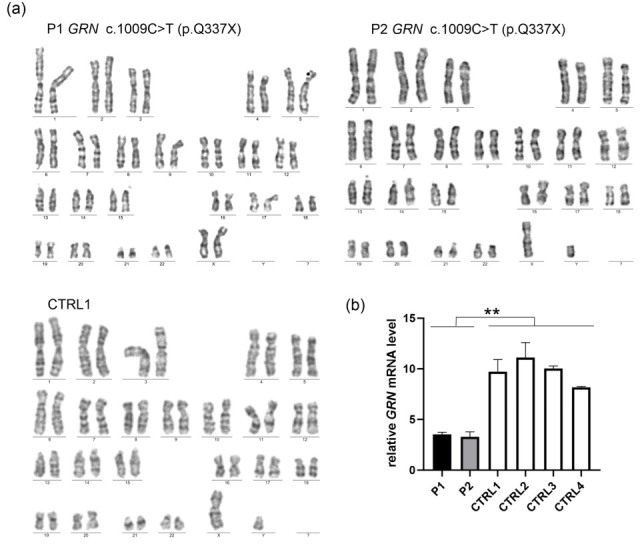
(**a**) The established iPSC lines—P1, P2, and CTRL1—exhibited a normal diploid karyotype (**b**) iPSCs derived from symptomatic (P1) and asymptomatic (P2) carriers of pathogenic *GRN* variant c.1009C>T (p.Q337X) had decreased mRNA *GRN* levels, compared to the lines obtained from a non-carrier with wild-type sequence (CTRL1) and other unrelated control iPSCs (CTRL2-CTRL4). *p* was calculated in a two-tailed *t*-test, ** *p* < 0.001.

Mycoplasma contamination was excluded in all iPSC samples (Table 1). No residual expression of Sendai virus-derived reprogramming vectors was detected by real-time PCR analysis, with early-passage iPSCs—still expressing viral sequences—serving as positive controls (Table 1 and Appendix A). Karyotype analysis by G-banding revealed normal chromosomal complements for all samples, with no detectable structural or numerical abnormalities (Figure 3a). Short tandem repeat (STR) profiling verified that the iPSC lines were genetically identical to the original donor fibroblasts, previously characterized by Gaweda-Walerych et al. [10]. Due to ethical considerations, STR profiling results are not disclosed to protect the personal information of cell line donors.

The presence of the pathogenic variant c.1009C>T was confirmed in iPSC lines derived from mutation carriers (P1 and P2), while the non-carrier line (CTRL1) retained the wild-type sequence, as determined by Sanger sequencing (Appendix A). Furthermore, iPSCs derived from both symptomatic (P1) and asymptomatic (P2) carriers of the *GRN* pathogenic variant c.1009C>T (p.Q337X) exhibited reduced *GRN* mRNA levels compared to the non-carrier line derived from a first-degree relative (CTRL1), and other unrelated control iPSCs (CTRL2-CTRL4) (Figure 3b).

### 2.2. Characterization of Neuroepithelial Stem Cells (NES) Lines from a Symptomatic Patient with GRN c.1009C>T (p.Q337X) Pathogenic Variant, Asymptomatic Carrier (P2) of Pathogenic GRN Variant c.1009C>T (p.Q337X), and a Healthy First-Degree Relative (CTRL1)

We have further generated NES lines from the established iPSC lines (P1, P2, CTRL1, and CTRL2—an unrelated control subject). For neural induction, the dual SMAD inhibition method was adapted using hNoggin (a BMP4 inhibitor), SB431542 (a TGFβ inhibitor), and CHIR99021 (a GSK3β inhibitor), as described before [14,15].

To confirm successful neural induction from iPSCs, NES cell lines were verified for the expression of protein markers such as Nestin and Sox2 by immunofluorescence (Figure 4a) [14]. In addition, the expression of other NES markers: *SOX1* (an early neural lineage marker indicating commitment to neuroectodermal fate), *PLAGL1* (associated with neural stem cell regulation), and *MKI67* (proliferation marker) was confirmed by RT-PCR, based on previously published data [14,15,16,17,18,19,20] (Figure 4b). For comparison in the RT-PCR analyses, and to highlight differences between the identity states of pluripotency, neuroepithelial stem cells, and terminal neuronal differentiation, we included one neuronal line designated CTRL5 NEU.

To generate this neuronal line, we used iPSC line IIMCBi002-A, derived from an unrelated control subject, which has been previously characterized in detail by Liszewska et al. [21]. We first used the IIMCBi002-A iPSC line to generate a NES line, which was subsequently differentiated into neuronal line CTRL5 NEU over a 6-week culture period [14,15] (see Section 4).

The mRNA expression of *SOX1, PLAGL1*, and *MKI67* in NES lines was compared to respective iPSC lines designated as P1 iPSC, P2 iPSC, and CTRL1 iPSC, and neuronal line CTRL5 NEU (Figure 4b). These juxtapositions clearly demonstrated the successful transition from pluripotency through neuroepithelial stem cell identity to terminal neuronal differentiation. As expected, *SOX1* mRNA level was the highest in NES, low in neuronal line, and almost absent in iPSCs (Figure 4b, left panel). *PLAGL1* expression was also low in iPSCs, while similar levels were observed in NES and neuronal line CTRL5 NEU (Figure 4b, middle panel) [15]. Consistent with its role in proliferation, *MKI67* mRNA expression levels were the highest in NES lines, slightly but significantly lower in iPSCs, and very low in the neuronal line (Figure 4b, right panel).

We also confirmed lower *GRN* mRNA levels in NES lines derived from carriers of *GRN* c.1009C>T (p.Q337X) pathogenic variant, compared to the control first-degree relative (CTRL1) and unrelated control NES line (CTRL2) (Figure 4c).

## 3. Discussion

In this study, we successfully generated iPSC lines from a family with pathogenic *GRN* variant c.1009C>T (p.Q337X), and further used them to obtain neuroepithelial stem cells. The generated iPSC and NES lines exhibited expression of markers specific to their respective developmental stages (Figure 1, Figure 2 and Figure 4). Moreover, as expected, *GRN* mRNA levels were markedly reduced in both iPSC and NES lines derived from mutation carriers compared to control cells (Figure 3 and Figure 4), consistent with previous analyses of dermal fibroblast lines from this family [10], and published data [3].

Immunostaining for four iPSC markers demonstrated nuclear localization of Nanog, Sox2, and Oct4, and membrane staining of TRA-1-81. These findings, together with quantitative analysis of *NANOG* and *OCT4* mRNA levels, confirmed successful reprogramming and maintenance of pluripotency (Figure 1). In addition, microscopic image analysis revealed the high purity (>90%) of the obtained iPSC cultures (Table 2). Embryoid body differentiation further confirmed the ability of the iPSCs to generate derivatives of the three germ layers. Likewise, the results presented in Figure 4b further support the proliferative status of our iPSC lines and distinguish them from NES, based on the mRNA expression profile of *MKI67*, a marker of cellular proliferation, and *SOX1*, a neural stem cell-specific marker. However, a limitation of our study is that we did not perform a genome-wide transcriptomic assay, such as PluriTest, which generates a pluripotency score—reflecting the degree of similarity to the transcriptomic signature of pluripotent stem cells based on comparison with extensive reference datasets—and a novelty score, which identifies abnormalities that may compromise oncogenic safety.

To characterize neuroepithelial stem cells, five markers were used: Nestin and Sox2 (immunofluorescence), and *SOX1*, *MKI67*, and *PLAGL1* (RT-PCR). The results of *SOX1, PLAGL1*, and *MKI67* mRNA expression levels in the obtained NES lines, compared to their respective iPSC lines, and a control neuronal cell line, align with previously published data [15,20,22,23,24]. Consistent with its role in early neural specification, *SOX1* was markedly upregulated in NES and the neuronal line CTRL5 NEU, while only residual expression was detected in iPSC lines (Figure 4b, left panel) [22,25]. This is in agreement with prior studies identifying *SOX1* as a neural stem/progenitor marker activated during early neuroectodermal differentiation [23,25]. Plag1 has been identified as a regulator of neurogenic potential in mouse neural progenitor cells [17,19,26]. *PLAGL1* expression, previously reported to be low in human embryonic stem cells (ESCs) and to increase upon differentiation into cortical neural cells [15,20], exhibited a similar pattern in our analysis. *PLAGL1* levels were low in iPSCs, but significantly elevated in NES and CTRL5 NEU lines (Figure 4b, middle panel), supporting its previously described role in neural lineage specification [20].

In line with its function as a proliferation marker, *MKI67* mRNA expression was highest in NES lines, moderate in iPSCs (but significantly lower than in NES), and very low in the post-mitotic neuronal line CTRL5 NEU (Figure 4b, right panel). This expression trend reflects the proliferative dynamics of NES cells during lineage commitment and expansion [23].

c.1009C>T pathogenic variant present in generated iPSCs and NES lines is considered rare among *GRN* mutations (between 1–4%) [7,8]. To date, among a cohort of 97 unrelated patients with frontotemporal lobar degeneration characterized by TAR DNA-binding protein 43-kDa-positive inclusions (FTLD-TDP), fifty distinct *GRN* mutations have been identified [7]. Notably, the c.1009C>T pathogenic variant was detected in two individuals in this study [7].

To date, neuronal lines have been generated from iPSCs derived from FTD patients with heterozygous and homozygous *GRN* pathogenic variants, including p.A9D, p. S116X, p.T272SfsX, p.R493X, IVS1 + 5G>C [27,28,29,30,31], reviewed in [3,32]. In addition, one organoid model has been developed based on iPSCs from FTD and NCL11 patients bearing the c.900_901dupGT *GRN* mutation [33]. Other human organoid models were based on *GRN* silencing, reviewed in [3]. To our knowledge, patient-based models with *GRN* c.1009C>T (p.Q337X) mutation have not been developed so far [3].

Taken together, our study provides the first patient-derived iPSC and NES models carrying the rare *GRN* c.1009C>T (p.Q337X) mutation. These models offer a valuable platform for investigating the molecular and cellular consequences of progranulin haploinsufficiency. Future differentiation into neuronal and glial subtypes, or the development of brain organoids, may reveal mechanisms underlying lysosomal dysfunction, neuroinflammation, and neurodegeneration in FTLD-TDP, thereby advancing our understanding of PGRN deficiency-related pathogenesis and guiding potential therapeutic strategies.

## 4. Materials and Methods

### 4.1. Reprogramming Fibroblasts into iPSCs

All cell cultures were maintained in a 37 °C incubator with 5% CO_2_. Primary skin fibroblasts were cultured as previously described [10,34]. Induced pluripotent stem cells (iPSCs) were generated by transducing 5 × 10^5^ fibroblasts using the Sendai virus-based CytoTune-iPS 2.0 Reprogramming Kit (Thermo Fisher Scientific, Waltham, MA, USA), following the manufacturer’s instructions. The Yamanaka factors (OSKM) were delivered at multiplicities of infection (MOI) of 5 (Oct4), 3 (Sox2), and 5 (Klf4/c-Myc). After approximately three weeks following viral transduction, colonies exhibiting typical iPSC morphology started to emerge. Individual colonies were manually picked and expanded in E8 medium (Thermo Fisher Scientific, Gibco, Waltham, MA, USA) on vitronectin-coated plates (Thermo Fisher Scientific). The cells were passaged every 3–4 days using 0.5 mM EDTA (Thermo Fisher Scientific, Invitrogen) with addition of 10 μM Y-27632 ROCK inhibitor (Enzo Biochem Inc., Farmingdale, NY, USA). The resulting iPSCs were validated for the expression of pluripotency markers by immunostaining and RT-PCR. Comprehensive characterization of each iPSC line was performed after passage 15. CTRL2–CTRL4 iPSC lines used in Figure 3b were generated from fibroblasts without *GRN* mutation derived from control individuals, unrelated to P1, P2, and CTRL1 subjects.

### 4.2. Real-Time-PCR (RT-PCR) Analysis

RNA was extracted using QIAzol Lysis Reagent (Qiagen, Manchester, UK) according to standard protocol, and subsequently reverse transcribed using the NG dART RT cDNA synthesis kit (EURx Molecular Biology Products, Gdansk, Poland; cat. no. E0801) with a mixture of random hexamer primers and oligo(dT) primers. Quantitative real-time PCR analysis was conducted with RT HS-PCR Mix SYBR (A&A BIOTECHNOLOGY, Gdansk, Poland) using a StepOne Plus system (Applied Biosystems, Foster City, CA, USA). Changes in gene expression were determined with the ∆Ct method using *GAPDH* levels for normalization, as described previously [35]. To confirm the elimination of the reprogramming vector, iPSC lines were analyzed for the presence of Sendai virus (SeV) genome, along with pluripotency markers *NANOG* and *OCT4*. Embryoid bodies obtained from P1, P2, and CTRL2 were analyzed for *TBXT* and *FOXA2*, mesodermal and endodermal markers, respectively. NES lines (P1, P2, CTRL1) were analyzed for the expression of NES-related markers (*SOX1*, *PLAGL1*, and *MKI67*), compared to respective iPSC lines. *GRN* mRNA levels were analyzed as described previously [34]. Primers [15,34,35,36] are listed in Appendix A.

### 4.3. Karyotype Analysis

Chromosomal analysis was performed in the Institute of Psychiatry and Neurology in Warsaw using G-banding staining. Over twenty metaphases were analyzed for each sample. Cytogenetic analysis and karyotyping were performed with a light microscope Nikon Eclipse 50i (Nikon Corporation, Tokyo, Japan) integrated with LUCIA Cytogenetics Imaging Software ver. 3.1. A detailed description of the procedure is provided in the Appendix A.

### 4.4. Sanger Sequencing and Authentication of iPSC Lines—Short Tandem Repeat (STR) Profiling

Sanger sequencing was performed on DNA extracted from iPSC lines to confirm the presence/absence of the c.1009C>T pathogenic variant, following previously published protocols [34]. To confirm that the obtained iPSC lines were genetically identical to the original donor fibroblasts, STR profiling was performed using the PowerPlex^®^ Fusion 6C System (Promega Corporation, Madison, WI, USA), which targets 27 STR loci plus Amelogenin. PCR products were electrophoresed on a 3500 Genetic Analyzer (Life Technologies Corporation, Carlsbad, CA, USA) and analyzed using GeneMapper ID-X software ver. 1.6 (Life Technologies Corporation, Carlsbad, CA, USA). STR profiling was done for the respective fibroblast–iPSC pairs in the Genomics Core Facility, Malopolska Centre of Biotechnology at Jagiellonian University, Gronostajowa 7A street, 30-387 Kraków. 14 loci recommended by the American National Standards Institute were reported in the final STR profiling report.

### 4.5. Mycoplasma Testing

The cell lines (fibroblasts and iPSCs) were mycoplasma-negative in the PCR Mycoplasma kit (MP Biomedicals LLC, Irvine, CA, USA).

### 4.6. Embryoid Body Formation

To confirm that the obtained iPSC lines can differentiate into the three germ layers, a modified “Embryoid Body (EB) formation using Essential 6™ Medium protocol” was applied (www.lifetechnologies.com/protocols, last accessed on 1 October 2025). Briefly, iPSCs were detached with EDTA and transferred into 6-well culture plate (Thermo Fisher Scientific, Gibco, Waltham, MA, USA) coated with Anti-Adherence Rinsing Solution (STEMCELL Technologies, catalog # 07010) and cultured in E8 medium with ROCK inhibitor for 24 h. From the second day, E6 medium was used and exchanged every other day. The cells formed three-dimensional floating aggregates, so-called embryoid bodies (EBs). After ten days in culture, the EBs were transferred onto 24-well cell culture plates containing coverslips pre-coated with matrigel (1:250 in E6 medium). The EBs attached to the matrigel surface and differentiated. After 10–14 days, cells were fixed with 4% paraformaldehyde and stained with germ layer-specific markers (Table 1) using an immunofluorescence procedure. RNA extraction from embryoid bodies (EBs) obtained from P1, P2, and CTRL2 was performed using QIAzol Lysis Reagent (Qiagen, Manchester, UK) according to the manufacturer’s protocol.

### 4.7. Generation of Neuroepithelial Stem Cells (NES) and Neuronal Differentiation

The obtained iPSCs (P1, P2, CTRL1, and CTRL2) were differentiated into neuroepithelial stem cells according to published protocols [14,37]. A detailed description of the procedure is provided in the Appendix A. NES identity was confirmed by the formation of rosette-like clusters and expression of protein markers such as Nestin and Sox2 was confirmed by immunofluorescence [14,37]. In addition, mRNA expression of NES markers, such as *SOX1*, *PLAGL1*, and *MKI67*, was analyzed by RT-PCR.

In this study, the iPSC line IIMCBi002-A, previously characterized in detail [21], was used. This iPSC line was derived from an unrelated control subject and was kindly provided by Dr. Ewa Liszewska from the Laboratory of Molecular and Cellular Neurobiology at the International Institute of Molecular and Cell Biology in Warsaw, Poland. We have generated NES line from the iPSC IIMCBi002-A line and it was subsequently differentiated into neurons over a 6-week culture period following the withdrawal of EGF and bFGF2, using Neurobasal medium supplemented with B27 (Thermo Fisher Scientific, Gibco) supplemented with Culture One supplement (Thermo Fisher Scientific, Gibco) during first week of differentiation [14,15]. The obtained neuronal line was designated as CTRL5 NEU, and was used in RT-PCR analysis of NES markers, *SOX1*, *PLAGL1*, and *MKI67* in Figure 4b.

### 4.8. Immunofluorescence

Cells were fixed in 4% formaldehyde for 10–15 min, washed three times with PBS, then permeabilized for 5 min. in 0.3% Triton X-100 in PBS, washed three times with PBS, and incubated for 1 h in blocking buffer comprised of PBS with 0.05% Tween-20, and 5% normal goat serum (NGS). Primary antibodies (Appendix A) were diluted in antibody dilution buffer consisting of PBS with 5% NGS, 0.05% Tween-20, and 0.02% sodium azide, then added to cells and incubated overnight at 4 °C. After washing three times, cells were incubated with diluted secondary antibodies for 1 h at room temperature in the dark. After washing three more times, nuclei were counterstained with DAPI and the coverslides were mounted on microscopy slides. Immunostained cells were imaged using fluorescent microscope Zeiss Axioexaminer.Z1 (Carl Zeiss Microscopy GmbH, Oberkochen, Germany).

### 4.9. Assessment of iPSC Line Purity

To evaluate the purity of the obtained iPSC lines, fluorescence images of Nanog- and DAPI-stained nuclei were acquired using a Zeiss Axio Examiner.Z1 microscope and analyzed with Cellpose (version 3.1.1.1) and ImageJ (version 1.54p). A detailed description of the procedure is provided in the Appendix A.

### 4.10. Statistical Analysis

For RT-PCR experiments, the relative values obtained from different iPSC or NES lines were used to calculate means, standard deviations, and statistical significance. The two-tailed unpaired *t*-test with Welch’s correction was used (GraphPad Prism 6.0). *p* < 0.05 was considered significant.

## 5. Conclusions

The iPSC and NES lines generated from a family with a pathogenic progranulin (*GRN*) variant c.1009C>T (p.Q337X) offer a valuable resource for studying early molecular changes, identifying therapeutic targets, and developing patient-specific models such as neurons, glia, and brain organoids. Our work contributes to a deeper understanding of PGRN deficiency-linked FTD pathogenesis and supports the development of precision medicine approaches for neurodegenerative diseases.

## Figures and Tables

**Figure 4 ijms-26-11242-f004:**
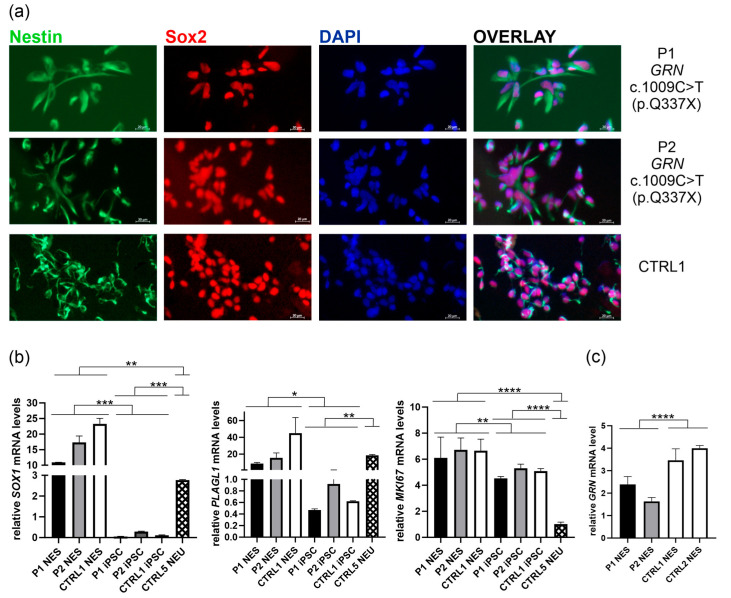
Characterization of neuroepithelial stem cells (NES) derived from the established iPSCs P1, P2, and CTRL1 (**a**) All lines express characteristic NES markers such as Nestin (green) and Sox2 (red); nuclei were stained with DAPI (blue); The upper panel: symptomatic patient (P1 carrying pathogenic *GRN* variant c.1009C>T (p.Q337X); The middle panel: asymptomatic carrier of pathogenic *GRN* variant c.1009C>T (p.Q337X) (P2); The lower panel: a non-carrier first-degree relative (CTRL1); Scale bar: 20 μm. (**b**) mRNA expression of NES-related markers (*SOX1, PLAGL1*, and *MKI67*) in the obtained NES lines (P1, P2, CTRL1), compared to respective iPSC lines, and a control neuronal line CTRL5 NEU; (**c**) lower *GRN* mRNA expression in P1 and P2 NES lines than CTRL1 and CTRL2 NES lines; P was calculated in two-tailed *t*-test, * *p* < 0.05, ** *p* < 0.001, *** *p* < 0.001, **** *p* < 0.0001.

**Figure 1 ijms-26-11242-f001:**
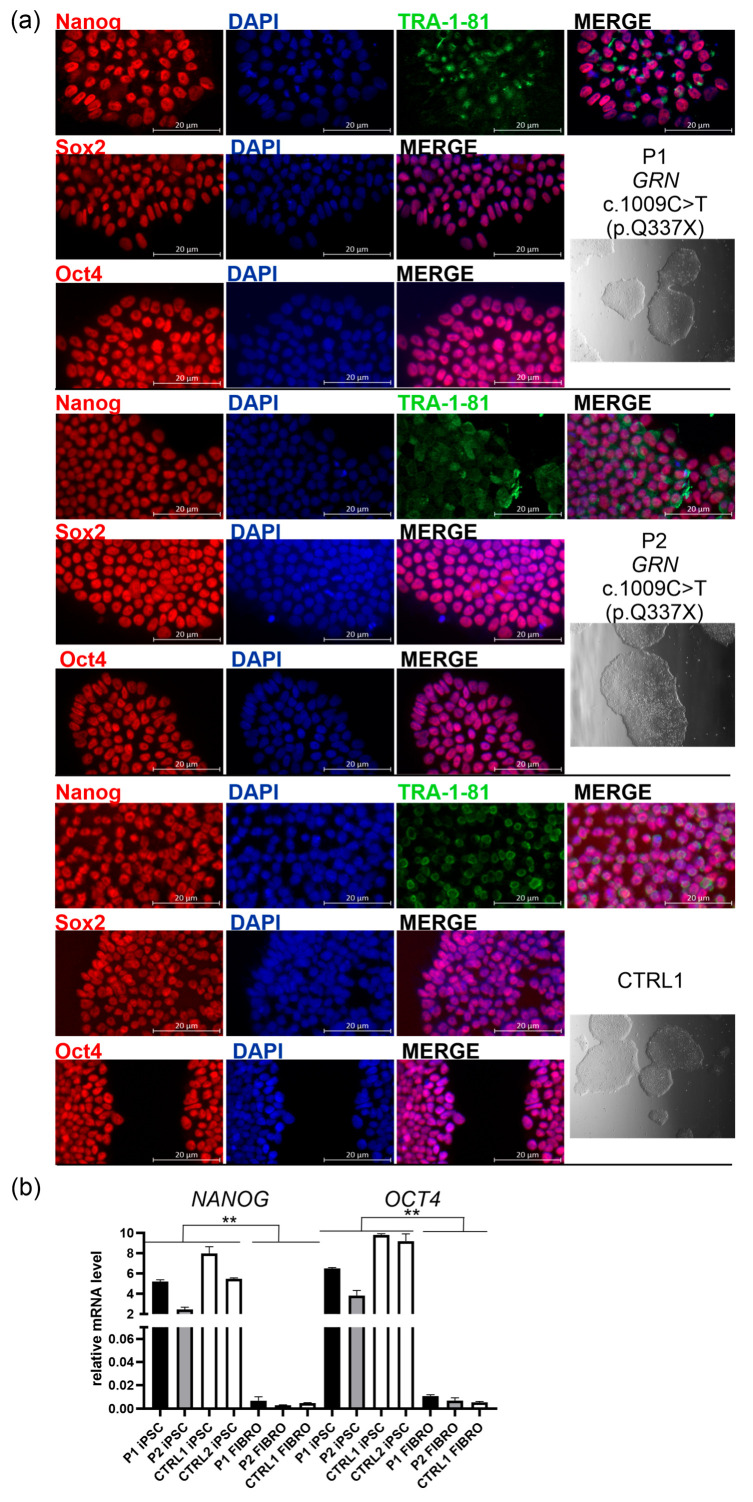
Characterization of induced pluripotent stem cell (iPSC) lines derived from two carriers of the pathogenic *GRN* variant c.1009C>T (p.Q337X): a symptomatic patient (P1), an asymptomatic carrier (P2), and a non-carrier first-degree relative (CTRL1); (**a**) right panel: all iPSC lines form colonies with well-defined boundaries, as visualized using Leica Integrated Modulation Contrast (IMC); greyscale images are shown below the corresponding cell line names; left panels: all iPSC lines showed positive immunostaining for pluripotency markers: Nanog or Sox2 or Oct4 (red), and TRA1-1-81 (green), and DAPI (blue). The figure displays split and merged fluorescence images acquired from either two (blue and red) or three channels: blue (DAPI), green (Alexa Fluor 488-conjugated secondary antibody), and red (Alexa Fluor 555-conjugated secondary antibody). Scale bar: 20 μm. The corresponding larger views, from which (**a**) images were cropped, are presented in Appendix A; (**b**) All iPSC lines: P1, P2, CTRL1 (non-carrier first-degree relative), CTRL2 (non-carrier unrelated control) express pluripotency markers, *NANOG* and *OCT4*, compared to respective fibroblast cells designated as P1 FIBRO, P2 FIBRO, and CTRL1 FIBRO; *p* was calculated in two-tailed *t*-test, ** *p* < 0.001.

**Table 1 ijms-26-11242-t001:** Characterization of the generated iPSC lines.

Category	Method	Result	Data
iPSC morphology	phase contrast microscopy	normal human induced pluripotent stem cell morphology	Figure 1a—right panel
iPSC phenotype	immunocytochemistry	positive immunostaining of pluripotency markers: Nanog, Oct4, Sox2, and TRA1-1-81	Figure 1a left panels, Appendix A
iPSC phenotype	real-time PCR	mRNA expression of pluripotency markers: *NANOG*, *OCT4*	Figure 1b
iPSC differentiation potential	Embroid body formation and STEMdiff ™ Trilineage Differentiation Kit	detection of the markers of the three germ layers, ectoderm (β3-Tubulin, Nestin), mesoderm (CXCR4), ectoderm/mesoderm (NCAM1), and endoderm (Sox17) by immunocytochemistry	Figure 2a, Appendix A
iPSC differentiation potential	real-time PCR	mRNA expression of the markers of mesoderm (*TBXT*), and endoderm (*FOXA2*)	Figure 2b
karyotype	G-banding	a normal karyotype was confirmed for the established iPSC lines, P1 (passage: 22, resolution: 400, total counted: 20), P2 (passage: 20, resolution: 400, total counted: 20), and C1 (passage 23, resolution: 400, total counted: 20)	Figure 3a
phenotype related to the *GRN* pathogenic variant	real-time PCR	mRNA expression of *GRN*	Figure 3b
virology	real-time PCR	no residual expression of Sendai virus-derived reprogramming vectors was detected in the established iPSC lines	Appendix A
mutation analysis	Sanger sequencing	confirmation of the presence of the pathogenic variant c.1009C>T in iPSC lines of carriers and the wild-type sequence in the non-carrier	Appendix A
iPSC lines authentication	STR profiling	complete concordance of the respective fibroblast—iPSC pairs for P1, P2, and CTRL1	not disclosed to protect the sensitive personal information of cell line donors

**Table 2 ijms-26-11242-t002:** Purity of the obtained iPSC lines P1, P2, and CTRL1 based on quantification of Nanog- and DAPI-positive nuclei.

iPSC Line	Percentage of Nanog- and DAPI-Positive Cells (%)	Regions of Interests (ROI) ^1^ Count
P1	96.2	18,113
P2	90.7	24,091
CTRL1	93.3	42,501

^1^ Each ROI corresponds to a segmented (algorithmically identified) nucleus.

## Data Availability

The data used in this study include STR profiling of human DNA samples, and thus cannot be openly shared due to personal data protection. The age of the subjects have been omitted to protect confidentiality and is available upon request.

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
