# Peer review of "Generation of Induced Pluripotent Stem Cells and Neuroepithelial Stem Cells from a Family with the Pathogenic Variant p.Q337X in Progranulin"

_ijms, 2025, doi:10.3390/ijms262311242_

Round 1
Reviewer 1 Report
Comments and Suggestions for Authors
The study describes the generation and characterization of induced pluripotent stem cells (iPSCs) and neuroepithelial stem cells (NES) derived from the fibroblasts of a family in which two members carry a pathogenic GRN mutation (c.1009C>T, p.Q337X, a known cause of frontotemporal dementia – FTD) (one symptomatic, one asymptomatic), as well as an unrelated control family member without the mutation. The iPSC and NES lines were assessed for pluripotency and lineage markers, differentiation potential, and genetic integrity. GRN mRNA expression was significantly reduced in the mutated cell lines. These new cell lines provide a patient-specific model system to study disease progression and early pathological changes, as well as for drug screening.
In general, the paper is written in a way that is easy to understand. The results are well structured and easy to follow.
The publication mainly describes the establishment of a cell line, which is typically published as a resource rather than as research. However, the authors differentiated the iPSCs into NES cells and found differences at the transcript level, which is why the study has not been classified as a resource.
The methods are described in great detail. This could be shortened.
Comments for improvement:
The characterization of the iPS cells is shown but incomplete.
Pluripotency is described using only three immunofluorescence stainings and two transcript analyses with very limited data. Usually, flow cytometry is performed to assess the purity of cell cultures. Additionally, further analyses are often conducted (digital microscopy, alkaline phosphatase, etc.). The figures are of rather low resolution. It is difficult to judge whether OCT4 and SOX2 are truly nuclear-localized, and the TRA-1-81 staining on the cell membrane is not clearly visible.
Genetic integrity is sufficiently described using a karyogram and G-banding. However, the specific karyotype is not mentioned; it is only stated that it is normal.
An analysis regarding the oncogenic potential is missing (e.g., Pluritest).
The differentiation potential is not well described using the immunofluorescence stainings. NCAM1 and CXCR4 are not typical mesoderm markers. CXCR4 is more often used as an endoderm marker. NCAM1 is also a marker of the neuroectoderm (it’s a synaptic protein). There are, in fact, other more established markers. For endoderm, only one marker was used; why not two?
The differentiation into NES is shown with only three markers. NESTIN and SOX2 by immunofluorescence (the figures show very different cell densities) and SOX1 by transcript analysis. Why was a different marker (SOX1 instead of SOX2 and NESTIN) used for quantification (transcript analysis)? Why was no flow cytometry performed? It is difficult to say whether differentiation was equally successful in all lines. The differences in GRN expression could result solely from this.
The selection of the different controls (“CTRL 5 NEU”) needs to be described more clearly, since the significant results are based on this comparison.
The choice of PLAGL1 in NES needs better justification.
The discussion is very brief and needs assessment of the results from the differentiation of iPSCs to NES in the context of the literature.
Author Response
|
Response to Reviewer 1
|
||
|
1. Summary |
|
|
|
We thank Reviewer1 for taking the time to review this manuscript. Please find the detailed responses below and the corresponding revisions/corrections highlighted in green in the re-submitted files.
|
||
|
2. Point-by-point response to Comments and Suggestions for Authors
|
||
|
Reviewer1: The study describes the generation and characterization of induced pluripotent stem cells (iPSCs) and neuroepithelial stem cells (NES) derived from the fibroblasts of a family in which two members carry a pathogenic GRN mutation (c.1009C>T, p.Q337X, a known cause of frontotemporal dementia – FTD) (one symptomatic, one asymptomatic), as well as an unrelated control family member without the mutation. The iPSC and NES lines were assessed for pluripotency and lineage markers, differentiation potential, and genetic integrity. GRN mRNA expression was significantly reduced in the mutated cell lines. These new cell lines provide a patient-specific model system to study disease progression and early pathological changes, as well as for drug screening. Comments 1: In general, the paper is written in a way that is easy to understand. The results are well structured and easy to follow. Response 1: We thank the Reviewer1 for this positive comment on our manuscript. Comments 2: The characterization of the iPS cells is shown but incomplete. Pluripotency is described using only three immunofluorescence stainings and two transcript analyses with very limited data. Usually, flow cytometry is performed to assess the purity of cell cultures. Additionally, further analyses are often conducted (digital microscopy, alkaline phosphatase, etc.). The figures are of rather low resolution. It is difficult to judge whether OCT4 and SOX2 are truly nuclear-localized, and the TRA-1-81 staining on the cell membrane is not clearly visible.
|
||
|
Response 2: We thank Reviewer1 for these insightful comments. We have substantially revised the figures. Immunofluorescence images were acquired using the Zeiss Axio Examiner.Z1 digital microscope equipped with high-resolution optics and Zen imaging software. This system supports precise localization of nuclear and membrane markers. In Figure 1, we now provide enlarged images of staining for four iPSC markers, namely Nanog, Sox2, and Oct4, and TRA-1-84. They clearly show nuclear staining for Nanog, Sox2, and Oct4, and membrane staining for TRA-1-84. In the supplementary data APPENDIX1 we provide Fig. S1a that displays the original, complete images. The regions outlined with a white dashed line were used to generate the enlarged images shown in Fig. 1a of the manuscript.
The Reviewer2 wrote that supplementary data was not available in the system for the first round of the review. We apologize for this oversight and now enclose APPENDIX1, as it contains important supporting data.
As to purity of our iPSC lines, we routinely monitored and manually purified the cultures using a gentle dissociation protocol with short EDTA incubation during passaging to get rid of differentiated cells.
In concordance with the Reviewer1 suggestion, we have included additional results based on the analysis of microscope images that demonstrate the high purity of our iPSC cultures as confirmed by (>90%) expression of core pluripotency marker Nanog (page 6 of the manuscript: description and Table 2 highlighted in green). More than 18000 cells were analyzed per line. Accordingly, we have updated Materials and Methods sections in the manuscript (page 14) and in the APPENDIX1 (page 8).
We would also like to draw Reviewer 1’s attention to the results presented in Figure 4b, which further support the pluripotent status of our iPSC lines. In this figure, RT-PCR analyses compare our iPSC lines (P1, P2, and CTRL1) with the NES lines derived from them. As expected, SOX1—a neural stem cell-specific marker indicative of early commitment to the neuroectodermal lineage—was expressed at high levels in NES lines, while only residual expression was detected in the iPSC lines, consistent with their undifferentiated state. Additionally, MKI67, a marker of cellular proliferation, showed high expression in both iPSCs and NES, reflecting the active proliferative capacity of these populations.
We appreciate the suggestion to include Alkaline Phosphatase (AP) staining. However, AP is primarily used as an early screening tool during the initial stages of reprogramming to identify emerging pluripotent colonies. In our study, we have confirmed pluripotency using a panel of four well-established markers — Nanog, Sox2, Oct4, and TRA-1-84 — which provide more specific and definitive evidence of successful reprogramming and maintenance of pluripotency. Including AP staining would be redundant in this context, as it does not offer additional specificity beyond the markers already presented.
|
||
|
Comments 3: Genetic integrity is sufficiently described using a karyogram and G-banding. However, the specific karyotype is not mentioned; it is only stated that it is normal.
|
||
|
Response 3: We have, accordingly precised the description of genetic integrity. “Karyotype analysis by G-banding revealed a normal chromosomal complement of 46, XX (female) for P1, 46, XY (male) for P2, and 46, XY (male) for CTRL1, with no detectable structural or numerical abnormalities.” (page 8)
Comments 4: An analysis regarding the oncogenic potential is missing (e.g., Pluritest).
Response 4: We acknowledge the Reviewer 1 suggestion regarding the assessment of oncogenic potential using tools such as PluriTest. While we did not perform a PluriTest or equivalent transcriptomic analysis in this study, we focused on validating pluripotency through a comprehensive panel of established markers (OCT4, NANOG, SOX2, TRA-1-81) and demonstrated stable morphology and high-purity cultures over extended passages.
Our iPSC lines were maintained under feeder-free, xeno-free conditions with careful passaging and manual removal of differentiated cells, minimizing the risk of transformation or spontaneous differentiation. Although we did not include a genome-wide expression assay in this study, we agree that such analyses could provide additional insights and will consider incorporating them in future work.
|
||
|
Comments 5: The differentiation potential is not well described using the immunofluorescence stainings. NCAM1 and CXCR4 are not typical mesoderm markers. CXCR4 is more often used as an endoderm marker. NCAM1 is also a marker of the neuroectoderm (it’s a synaptic protein). There are, in fact, other more established markers. For endoderm, only one marker was used; why not two? Response 5: We thank the Reviewer1 for this insightful comments. Due to financial restrictions of this project we could use only a limited number of antibodies. According to the Reviewer1 suggestions we have performed additional analyses using cDNA retrotranscribed from RNA obtained from embryoid bodies (EB). Results from embryoid bodies are now presented in revised Figure 2b. We have included additional RT-PCR analyses for two germ layer markers: Brachyury (mesoderm) and FOXA2 (endoderm), to further support the differentiation capacity of our iPSC lines (Figure 2b). Description of the results from Fig2b is on page 8, and in caption for Fig 2, and has been added to Table 1. We have updated Material and Methods section on Page 13 (Embryoid body formation). Regarding mesodermal markers, we have in parallel assessed the differentiation potential of P1, P2, and CTRL1 iPSC lines into the three germ layers using the STEMdiff™ Trilineage Differentiation Kit (STEMCELL Technologies). To confirm differentiation of P1, P2, and CTRL1 iPSC lines into a homogeneous mesodermal population, CXCR4 staining was performed as recommended in the protocol of the STEMdiff™ Trilineage Differentiation Kit. CXCR4 staining revealed both membrane-bound and cytoplasmic localization (the supplementary material APPENDIX1 Fig S3). Regarding SOX17, we have added a new paragraph in the Results section (page 6) to emphasize that among endodermal markers, SOX17 is considered the most specific and reliable indicator of definitive endoderm in embryoid bodies derived from iPSCs. Comments 6: The methods are described in great detail. This could be shortened. Response 6: We appreciate the Reviewer1 feedback regarding the level of detail in the Methods section. Our intention was to provide a comprehensive and transparent account of the experimental procedures to ensure reproducibility and clarity for future studies. According to the Reviewer1 suggestion we have moved detailed descriptions to the supplementary material APPENDIX1 (page 7 and 8, highlighted in green). Comments 7: The differentiation into NES is shown with only three markers. NESTIN and SOX2 by immunofluorescence (the figures show very different cell densities) and SOX1 by transcript analysis. Why was a different marker (SOX1 instead of SOX2 and NESTIN) used for quantification (transcript analysis)? Why was no flow cytometry performed? It is difficult to say whether differentiation was equally successful in all lines. The differences in GRN expression could result solely from this. Response 7: We thank Reviewer 1 for these comments and would like to clarify that the differentiation into NES was assessed using a broader panel of markers than initially noted. In addition to Nestin and Sox2 (immunofluorescence) and SOX1 (transcript analysis), we also evaluated MKI67 (proliferation marker) and PLAGL1 (neural lineage-associated transcription factor) by RT-PCR, as shown in Figure 4b. These markers collectively support the identity and proliferative status of the NES lines. We have added a dedicated paragraph in the Discussion section explaining how SOX1, MKI67, and PLAGL1 confirm NES identity (page 11 highlighted in green), along with appropriate references. Regarding the use of SOX1 for transcript-level quantification, our intention was to complement the immunofluorescence data with an early neuroectodermal marker. SOX1 is a well-established marker of neural progenitors at the mRNA level. We chose SOX1 for RT-PCR analysis to avoid redundancy with SOX2, and to provide an additional validation beyond Nestin and Sox2 protein expression. Regarding the differences in cell density observed in the immunofluorescence images in Figure 4a (updated numbering), they were due to uneven plating on slides. We agree that this could be confusing to readers, and we have now provided representative cropped images for NES lines in Figure 4a. We appreciate the Reviewer1 suggestion regarding flow cytometry. In our study, immunofluorescence analysis revealed that NES cells were positive for Nestin and Sox2, with consistent and homogeneous staining across the cultures. We used the Calvo Garrido et al. 2021 STAR Protocol for differentiating iPSCs into neuroepithelial stem cells (NES), that provides high efficiency and purity of NES marker expression. This protocol recommends verifying the expression of the NES cell markers SOX2 and NESTIN by immunohistochemistry, FACS or qPCR. We have performed verification by immunohistochemistry and qPCR. Given uniform expression observed in immunostaining, the clear marker distribution and cell morphology, we did not consider flow cytometry essential for further validation. Nonetheless, we acknowledge that flow cytometry is an excellent method to analyze the purity of iPSC and NES lines. We would like to clarify that the observed differences in GRN expression are not indicative of unequal differentiation efficiency across NES lines. All lines were differentiated under identical conditions, and we have now corrected imaging bias due to uneven plating on slides. As shown in our immunofluorescence data, all NES cultures consistently expressed Nestin and Sox2, confirming successful neural induction. The variation in GRN levels stems from the fact that two of the NES lines (P1 and P2) carry pathogenic GRN mutations, which affect GRN transcript level. Comments 8:The selection of the different controls (“CTRL 5 NEU”) needs to be described more clearly, since the significant results are based on this comparison. Response 8: We thank Reviewer1 for highlighting this important point. To clarify the selection of the neuronal control line “CTRL 5 NEU,” we have added a detailed description in Results Section 2.2 (page 9, highlighted in green), and in Materials and Methods, section entitled “Generation of neuroepithelial stem cells (NES) and neuronal differentiation” (page 13). We have included this neuronal line to highlight differences between the identity states of pluripotency (iPSC), neuroepithelial stem cells, and terminal neuronal differentiation, which are critical for interpreting the significant results observed in Fig. 4b. We have also included the appropriate reference. We have also corrected an error in this section. In the previous version, we mistakenly written that RT-PCR comparisons were made between NES and fibroblast lines: probably copy-paste error). This sentence has now been corrected (page 9, highlighted in grey): “The mRNA expression of SOX1, PLAGL1, and MKI67 in NES lines was compared to respective iPSC lines designated as P1 iPSC, P2 iPSC, and CTRL1 iPSC, and neuronal line derived from unrelated control designated as CTRL5 NEU”. Moreover, we have clarified the provenience of CTRL2–CTRL4 iPSC lines used in Fig. 3b in Materials and Methods, section Reprogramming fibroblasts into iPSCs. Comments 9: The choice of PLAGL1 in NES needs better justification. Response 9: We appreciate the Reviewer1 suggestion to justify the choice of PLAGL1. We would like to clarify that PLAGL1 was selected based on its emerging role in early neuroectodermal development. Recent studies have identified PLAGL1 as a transcriptional regulator involved in neural progenitor cell fate decisions and in modulating gene networks associated with neurogenesis. We have added a dedicated paragraph discussing PLAGL1’s role in the Discussion section (page 11 highlighted in green), along with appropriate references.
Comments 10: The discussion is very brief and needs assessment of the results from the differentiation of iPSCs to NES in the context of the literature. Response 10: We have addressed the Reviewer1 suggestion by expanding the Discussion section. A dedicated paragraph (highlighted in green, page 11) has been added to discuss the results of iPSC differentiation into NES cells in the context of existing literature. This includes a comparative analysis of mRNA levels of SOX1, PLAGL1, and MKI67, and Nestin and Sox2 immunostaining. This paragraph highlights how our findings align with previously reported profiles of neural stem cell identity. Relevant references have been included. |
||
Reviewer 2 Report
Comments and Suggestions for Authors
In this study, the authors presented the generation of iPSCs harboring p.Q337X mutation in progranulin, along with non-carrier control. The quality of the iPSCs was assessed through standard pluripotency assays, and the lower expression of GRN was confirmed.
Major comments:
- In Figure 3a, during the generation of NES, why did the CTRL1 group have much higher cell density than P1 and P2, considering the same amount of 200,000 iPSCs were seeded on Day 0? Was there cell loss during this process? And how does that link to the role of GRN in neurons?
Minor comments:
- In Figure 1b, please adjust the breaks on the Y-axis so that the P2 iPSC values are shown.
Other comments:
- I’m not able to see the supplementary files in the system. Please make sure they are available for review.
Author Response
Response to Reviewer 2
|
1. Summary |
|
|
|
We thank Reviewer2 for taking the time to review this manuscript. Please find the detailed responses below and the corresponding revisions/corrections highlighted in green in the re-submitted files.
We thank Reviewer2 for the positive assessment of the introduction, research design, methods description, results presentation, and conclusions.
|
||
|
2. Point-by-point response to Comments and Suggestions for Authors
|
||
In this study, the authors presented the generation of iPSCs harboring p.Q337X mutation in progranulin, along with non-carrier control. The quality of the iPSCs was assessed through standard pluripotency assays, and the lower expression of GRN was confirmed.
Comments 1: In Figure 3a, during the generation of NES, why did the CTRL1 group have much higher cell density than P1 and P2, considering the same amount of 200,000 iPSCs were seeded on Day 0? Was there cell loss during this process? And how does that link to the role of GRN in neurons?
Response 1: We thank Reviewer 2 for raising this point. The apparent differences in NES cell density observed in Figure 4a (updated numbering) were due to uneven plating on slides rather than cell loss during differentiation. We agree that this could be confusing to readers, and we have now included representative cropped images of NES lines in Figure 4a. We would also like to emphasize that we carefully examined any potential differences that might arise from carrying GRN mutations.
Comments 2: In Figure 1b, please adjust the breaks on the Y-axis so that the P2 iPSC values are shown.
Response 2: We have made the corrections according to the Reviewer2 suggestions.
Comments 3: I’m not able to see the supplementary files in the system. Please make sure they are available for review.
Response 3: We apologize for this oversight and thank the Reviewer2 for bringing it to our attention. We have now ensured that APPENDIX 1 is properly included with the revised submission. We were convinced that APPENDIX 1 had been successfully uploaded in the initial submission as it contains important supporting data.